# A piece of a puzzle—Patient and psychologist experiences of the Affect School as additional treatment in a Swedish eating disorder unit

**Suzanne Petersson**[1,2]*, **Ingrid Wåhlin**[3,4]

**1** Department of Medicine and Optometry, Linnaeus University, Kalmar, Sweden, **2** Division of Rehabilitation, Region Kalmar County, Kalmar, Sweden, **3** Department of Health and Caring Science, Linnaeus University, Kalmar, Sweden, **4** The Research Section, Region Kalmar County, Kalmar, Sweden

☯ These authors contributed equally to this work.
* suzanne.petersson@regionkalmar.se

**Data Availability Statement:** The raw data necessary to replicate this study consist of participant interview transcripts. Participants in this study did not consent to have their full transcripts

## Abstract

### Background

Emotion dysregulation has been shown to be a transdiagnostic characteristic of eating disorders. The Affect School aims to enhance emotional awareness and the ability to perceive and express emotions. This study was conducted as part of an RCT where patients with various eating disorders were randomised to participation in the Affect School as a supplement to treatment.

### Aim

To explore Affect School participants' and leaders' experiences of the Affect school at an Eating Disorder outpatient unit in Sweden.

### Method

Nine patients with eating disorder diagnoses and three Affect School leaders at an eating disorder outpatient clinic in Sweden were interviewed for their experiences of the intervention. The interviews were analysed with Thematic Analysis.

### Result

Eight themes were revealed at the analysis, five from the participants: "Worries about group participation", "Not alone anymore and gaining new insights about oneself", "Shared stories can also be painful", "Relationships outside the Affect School as a foundation for affective work", and "A change is coming", and three from the leaders:"Affect awareness is important in eating disorders", "Group meetings create opportunities and challenges", and "The Affect School setup needs more customisation".

### Conclusion

The results suggested that the Affect School provided an acceptance for experiencing all sorts of affects. Both leaders and participants considered working with affects necessary,

made publicly available, and the issue of publishing interview transcripts has not been examined by the ethics review (by the Research Ethics Committee of Linköping;2017/531-31), therefore this disclosure would contravene the terms of their consent. Such restrictions have been imposed by the Region Kalmar County, who as principal for this study, is chiefly responsible for the research data collected. Data are available at the Unit for research and development at Region Kalmar län (contactforskningssektionen@regionkalmar.se, or phone: +46 480-426684) for researchers who meet the criteria for access to confidential data.

**Funding:** SP received a grant from Fredrik and Ingrid Thurings' Foundation (in Swedish: Fredrik och Ingrid Thurings Stiftelse) grant No 2017-00308 but the funders had no role in study design, data collection and analysis, decision to publish, or preparation of the manuscript. No other fundations were received.

**Competing interests:** The authors have declared that no competing interests exist.

**Abbreviations:** ABC, AnorexiBulimiCenter (psychiatric outpatient clinic for treatment of eating disorders in Sweden); AN, Anorexia Nervosa; AS, Affect School; BN, Bulimia Nervosa; DSM-5, Diagnostic and Statistical Manual of Mental Disorders, 5th edition; ED, Eating Disorder; RCT, Randomized Controlled Trial; SEDI, Structured Eating Disorder Interview; TA, Thematic Analysis.

although participants reported no changes in their life situation or condition after the intervention but considered the intervention as part of a process with gradual results. Being part of a group and sharing experiences was experienced as positive by participants and leaders, but the model could be further developed to better adapt to patients with an eating disorder.

## Introduction

Emotion dysregulation has been shown to be a transdiagnostic characteristic of eating disorders, ED, and the higher the degree of emotion dysregulation the higher degree of ED symptomatology [1]. It has been suggested that deficits in emotion regulation are predisposing factors for the development of ED [2, 3]. Fox [4] described how a sample of patients with anorexia nervosa (AN) showed poor meta-emotional skills, but also that they could acknowledge emotions but were uncertain whether they were permitted to do so, and thus suppressed negative emotions, especially anger, to avoid rejection. Problems with suppressing, and thus understanding and coping with emotions, lead to difficulties in everyday life, not least interpersonally [5]. The concepts of emotions and affects differ, affects can be defined as very brief, innate biological responses to stimuli, manifested in bodily, above all facial, autonomic responses, serving as "motivational fuel", while emotions are defined as a blend of affects along with psychological and biographical experiences [6]. A meta-study of depression treatment showed that by adding emotion regulation skills to cognitive behavioural therapy the effects of the treatment increased [7]. Different psychotherapeutic interventions aiming to encourage experiencing and expressing emotions have been used in treatment for patients with AN with acceptable and efficacious results [8]. Emotion-focused group therapy has also been evaluated for patients with bulimia nervosa (BN) [9].

The Affect School, AS, originates from the Department of Psychology at Umeå University, Sweden. The initial aim was to develop a psychoeducational intervention to increase affect awareness for psychosomatic problems e.g. pain, [10]. Chronic pain has been associated with emotion dysregulation, thus the AS aimed at increasing emotional awareness and the ability to perceive and express emotions [11]. The method is based on the affect theories developed by Tomkins, Nathanson, and Ekman [10]. Since its inception the AS has been used within a number of areas in conjunction with pain treatment, e.g. stress-related problems, depression, and anxiety [11–14].

The AS is manualised and training under supervision is required to become an AS leader. The manual may not be distributed to persons without training in the method [10]. Only smaller deviations from the manual are accepted, and the theoretical section on pain could be replaced by a corresponding section on EDs, without changing the name of the intervention. Thus, the present study was approved by the author of the AS, Professor B-Å Armelius (personal communication 2017-09-19). The intervention consisted of an eight-session group treatment with two-hour weekly sessions [10, 12]. Participants received handouts at the beginning of each session to follow the theoretical part of the session. All sessions followed the same approach: initially 30–60 minutes' psychoeducation about the affect of the session, followed by reflections and a break. After the break a discussion followed on the specific affect, and each group participant was encouraged to talk about an incident that had started this affect for her/him. This was eventually followed by a longer reflection on the subject. Participants were not informed of which affect was to be discussed at the following session but would discover the

theme at the start. The affects that were discussed were joy, fear, interest/startle, shame, anger, dissmell/disgust, and worry. The last session in the original layout was devoted to stress, pain, and its prevention [10, 12]. For this study the last session was modified, taking up ED to match participants. Originally, the set of group sessions could be followed by individual script analyses, however this was not the case in this study.

There has been substantial research on eating disorders and emotion regulation, although many questions remain unanswered. As far as we know, studies on AS interventions for patients with ED are scarce. The aim of this study was to describe patient and therapist experiences of affect school according to eating disorders.

## Methods

### Study setting

The study was conducted at a youth/adult, integrated psychiatric outpatient clinic specialising in the treatment of ED in southern Sweden. The clinic receives approximately 150 new patients yearly.

### Diagnostic evaluation of eating disorder diagnoses

Diagnostic evaluation was performed by experienced psychologists or psychotherapists at the clinic using clinical interviews based on the Structured Eating Disorder Interview (SEDI; adapted to the DSM-5 criteria) [15].

### Participants

With a start in January 2018 and continuing to December 2019, 46 patients, all women ($\geq$18 years) with different ED diagnoses at the clinic, agreed to participate in an RCT aimed at evaluating the AS as a supplementary intervention for patients with ED (listed on the ISRCTN registry with study ID ISRCTN11278582). Six participants dropped out. Twenty-one participated in the intervention while the remainder (n = 19) formed a control group. Parallel to the intervention both groups received treatment as usual which in these cases included individual psychotherapy (CBT) and in some cases also day care and/or physiotherapy. After completed participation in the AS participants were asked to be interviewed about their experiences. Nine participants from five AS groups agreed to participate, ages ranging from 19–43 years of age (Mean = 26.6; Median = 25). The distribution of the DSM-5 eating disorder diagnoses was: two participants with AN, one with BN, and six participants diagnosed with Other Specified Feeding or Eating Disorder (including five participants diagnosed with atypical AN, and one with low frequency BN). Participants had ED with long durations and the self-estimated ED duration at the time for the study was between six months to nine years (Median = 8). Three participants reported having earned a university degree, one was at university, four had completed grammar school, and one had completed an elementary school education. None were unemployed, but four were on sick leave at the time of inclusion. The AS groups were small (three to five persons).

### Affect school leaders

The AS was introduced by a master's level psychologist at the ABC. After a pilot study three women master's level psychologists at the unit led the AS groups evaluated in the present study. The psychologists worked in pairs, and one followed all groups together with one of the others. Two of the psychologists had been working with ED treatment for roughly one year before the start of the study, and one started with the AS directly after completing her training

at the university. Their ages varied from 25 to 32. All AS leaders agreed to participate in an interview.

## Ethical approval and consent to participate

The study was conducted according to the principals of the Helsinki declaration. All participants were adults ($\geq$18 years old). Prior to interviews participants were provided written and oral information about the study. Participants signed an informed consent to participation which was attached to the written information. It was emphasised that participation was voluntary, and that participation or refusal would not affect treatment. Participants were informed that presentation of the data would be handled with confidentiality so that no statement could be traced to any single informant. The study was approved by the research ethics committee of Linköping (2017/531-31). The interviewers had no treatment relationship to the interviewees.

## Data collection

Data was collected through semi-structured interviews. The interviews were based on two guides constructed by the authors, one for participants and one for AS leaders (see Figs 1 & 2). The authors had no relation to the participants in the AS, and the interviews of the three AS leaders were conducted by the second author who only met them at the time of the interview. Open questions were followed by general probing questions which aimed to encourage richer narratives [16]. The participants' interviews started with broader questions concerning their personal experience of the intervention. The questions then became more specific focusing on experiences of emotions. The latter part of the interview also included questions about possible new knowledge, affect awareness, and connections between emotions and ED. Four pilot

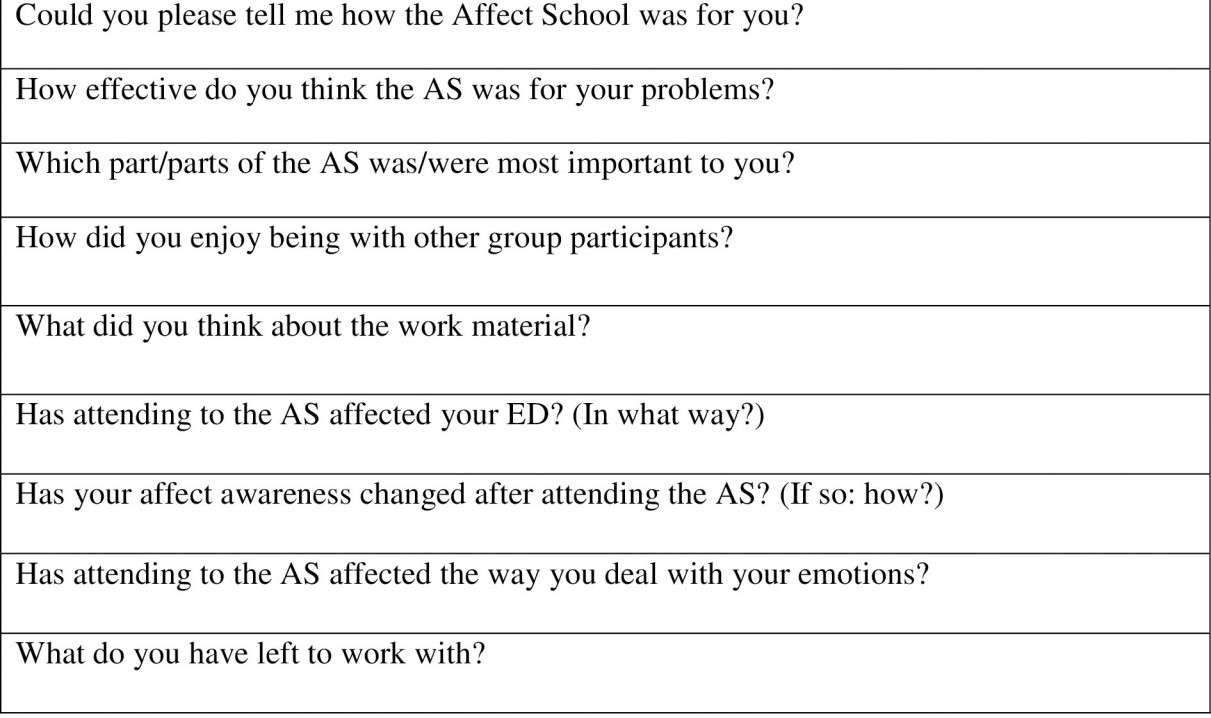

| Could you please tell me how the Affect School was for you? |
| --- |
| How effective do you think the AS was for your problems? |
| Which part/parts of the AS was/were most important to you? |
| How did you enjoy being with other group participants? |
| What did you think about the work material? |
| Has attending to the AS affected your ED? (In what way?) |
| Has your affect awareness changed after attending the AS? (If so: how?) |
| Has attending to the AS affected the way you deal with your emotions? |
| What do you have left to work with? |

**Fig 1. Interview questions (participants).**

| What has it been like to lead the AS? |
| --- |
| What do you think about the group format? |
| What do you think about the work material? |
| Participants had different ED diagnoses, how has this affected the groups? |
| How do you view the AS as an adjunctive treatment for patients with ED? (Please develop, for whom would it be beneficial?) |
| What would you like to change? (Why?) |
| Have you other thoughts related to the intervention? |

**Fig 2. Interview questions (AS leaders).**

interviews were conducted in order to test the interview guide, and it was decided to change the order of the first two questions. After revision the first question was: "Could you please tell me about your experience of the Affect School?" (Fig 1). Pilot interviews were later included in the study.

The interviews with the AS leaders started with questions about how it was to lead the AS, about the material and group composition. The later part provided an opportunity to relate the development of the intervention and other views of the AS as additional treatment for eating disorders (Fig 2).

Interviews were audio recorded and lasted between 33 and 52 minutes. Both authors conducted interviews with participants, IW interviewed all AS leaders.

## Data analysis

Data was analysed by inductive thematic analysis [17]. Thematic analysis was considered suitable because of its theoretical flexibility and descriptive function of divergences and convergences of participants' experiences.

Data was transcribed verbatim by the first author and a research colleague from another project. Text analyses were conducted by both authors and analysed as follows:

1. Both authors read and reread all interviews individually. Sentences and parts of sentences were recognized as initial codes and marked in the transcripts.

2. Preliminary codes were assigned in order to describe the content.

3. Patterns or themes were sought across codes from all interviews. Large paper sheets, post-it notes and pens with different colours were used at this stage.

4. The authors met and discussed the themes and structure of the results. Participants' and AS leaders' narratives were analysed separately but were compared in the Discussion section

later. Themes were reviewed, and the material was divided into six themes for participants and three themes for the AS leaders.

5.  Themes were named in a collaboration between the authors.

6.  Illustrative quotations were chosen by the first author followed by discussions with the second author. The themes were discussed from a theoretical perspective and the report was then compiled by both authors.

During this process, the number of themes was reduced and some were renamed. The theme system was then found consistent with the data and therefore regarded as final. The first author has been working with ED treatment for 20 years as a licensed psychologist and psychotherapist. The second author is an experienced registered nurse without prior knowledge of ED but with extensive experience of interview studies. In the quotations, the authors have sometimes filled in omitted words for increased readability. These are within brackets.

## Results

The following eight themes were revealed during the analyses of the interviews, the first five from the participants: "Worries about group participation", "Not alone anymore and gaining new insights about oneself", "Shared stories can also be painful", "Relationships outside the Affect School as a foundation for affective work", and "A change is coming", and three from the leaders:"Affect awareness is important in eating disorders", "Group meetings create opportunities and challenges", and "The Affect School setup needs more customisation".

### Affect school participant experiences

**Worries about group participation.**   Prior to and at the start of the AS, many participants were worried about the group format. They were not used to voicing thoughts and feelings, and especially not with strangers. This made them fearful of the first meeting, and sometimes even before every new meeting, as they did not know what would be discussed. The coming eight weeks' treatment could then be experienced as a lifetime.

*"I don't like speaking in front of people. . .it wasn't what I expected, but on the other hand thought it would. . . feel tougher, that I would just be nervous, that is, nervous mostly since it's so difficult to meet new people, and then all of a sudden just let out what you feel so it gets very deep, talking about what you don't want to talk about with just anyone, then you just go even further, I thought that was tough"* (7).

*"So the first time I was terrified when they said to* (laughs) *say something like. . ., yeah just me talking, no discussion, but that they would ask me questions, and I just felt "no!" "what have I gotten myself into?"* (40).

### Not alone anymore

After the first one or two sessions participants found themselves to be "shockingly comfortable" in the group. Some recognized coming further than expected in their work with the ED. Regardless of perceived similarities or differences with group members the group experience was described as a large recognition factor; participants did not feel as bad as before, they learned from each other and were helped by or helped each other. They no longer felt alone with their thoughts.

*"You felt as if everyone opened up and could share that, yes, I agree, I feel the same way and don't feel so alone. That. . . it's normal that we perhaps, in that group, have a bit more extreme thoughts, or that most of them there didn't feel so well. But an understanding was gained of not being alone, and that there are others that, yeah really, can feel differently too, or similarly, and that was kind of important, and you understood your own feelings a bit better"* (29).

*"You often feel pretty lonely in your chaos, and feel no one understands you, but here were others that could speak of things similar to your own experiences"* (23).

Other people's stories opened up new ways of thinking, thus providing insights into new ways of dealing with different situations. Listening to others could also make it easier to put your own feelings and experiences into words. The insight into affects made the participants reflect to a greater extent on what they actually felt in different situations.

*"That was probably really good since I was then truly forced to reflect and become aware even in everyday life afterwards. When feelings arise you think a bit, what am I really feeling now, what's happened and what am I feeling, and so on. Then it's still a bit chaotic inside but I've gained an idea of how you can reflect a bit on it"* (23).

By understanding their emotions better, and also gaining acceptance for them, participants could reflect upon their difficulties experiencing negative emotions and gain some new knowledge as well that could be related in ways which had previously been perceived from other perspectives. Such insights could also include ideas of what a certain affect means. Shame, for example, does not necessarily mean that you feel ashamed in front of others but that you can be ashamed of yourself.

*"I've always been like "I'm not ashamed of anything"*! *But when you realise that maybe it's not so embarrassing or shaming for making a fool of myself. But then, when getting into deeper water, I might not feel ashamed in front of others but feel ashamed within myself instead, and then get it that shame is not just screwing up in front of others but (something that) runs deeper, and that was tough. I hadn't thought about it like that before"* (7).

## Shared stories can also be painful

Despite the experience of support and connection within the group, some participants reported obstacles regarding sharing experiences of ED behaviour. Some reported that the intervention stirred up many emotions, and that it was impossible to take in so much at one time. One participant experienced that listening to another group member's story of how she starved herself or exercised in order to lose weight was triggering, even though she had gotten over such behaviours.

*"All of us girls sitting there had some kind of eating disorder, and I don't think it should be presented in a triggering way. I don't want to hear how many days someone has starved themselves. I think that's unnecessary information, so I was very upset and didn't intend to go back, because I don't think you should talk about that"* (9).

Even if participants were positive toward the possibilities of sharing experiences, supporting and helping each other, stories about suffering or negative life-events could be experienced as

almost overwhelming and difficult to handle. A few participants described other participants' stories as painful to listen to, and also told how they usually took on other's suffering.

*"I thought it interesting to listen even though it was awful to listen to some things, ugh! I have a habit of taking on other's burdens, wallowing in other's problems, and feel that ah. . .it would be highly burdensome for me and perhaps I should let it go, let it go as fast as possible"* (7).

### Relationships outside the Affect School as a foundation for affective work

Relationships were considered crucial for the development of affect awareness by the participants. By noting shared experiences a change was possible. The group format made this possible to reflect on. In addition, half of the participants worked with the AS material between sessions and reported discussing the content with family members, their psychologist (with whom some of them had weekly or bi-weekly sessions), or as one who practiced with her clients at work told:

*"I work in healthcare, so I'm really helped by these different feelings. By meeting others and better understanding oneself, you can better understand others as well"* (9).

*"I discussed things you maybe don't ordinarily talk about with my partner and my mother. It would be like if I, maybe, shared it* (about the handouts) *. . .it would be good if my mom and I did it together, if we would gather from our experiences and have it as a starting point"* (41).

### A piece of a puzzle–a change is coming

All participants were positive about their participation in the AS. They described that they, thanks to the intervention, understood that all affects play an important role, and could even accept the presence of negative ones. Talking about one's emotions became de-dramatised. Many felt that they should not bother others by expressing anger or sadness. By appearing carefree and happy they spared others, or some experienced carefreeness as a way of gaining approval. Being aware that it was natural to be sad at times gave relief for not having to feign happiness a considerable part of the time. They also described how they learned to differentiate between affects, and to name occurrences in the body in different emotional states.

*"Now I don't feel it's the food's fault when I get these impulses "now I have to throw up", but why are you going to throw up? It's. . .not the food that's done something to you but a feeling that's lying in wait!"* (30).

*"It was also a slight, an aha moment that, yeah, everyone has a function, maybe there is no function but some sort of survival function or, maybe it's no longer relevant but, if you think about it you can more easily understand where it comes from"* (41).

*"It's hard to start feeling better if you don't know what makes you feel bad"* (29).

Despite this, participants reported that the intervention had neither helped them with their mood, nor led to significant changes in their lives or behaviours. Their overall picture was that there would be a change, but it would come later, and that the intervention was "short in relation to an entire lifetime":

*" Well, I feel that it's helped me in that, in that it'll help me, sort of. Or I think maybe it'll help me. But just now maybe it was just a step that I, yeah, talked about it with some others and*

*got to hear a little of what they thought and felt. And, sort of, that you didn't feel so lonely with different emotions and such. So that's what sort of helped me most, to just talk and be open, I feel. But then, to process and maybe think, and sort your feelings and feel it'll get easier, yeah, later"* (29).

*"I feel like Affect School was maybe, yeah, the first pieces of the puzzle I had to piece together. It was, like, the start I had to understand, to find out why I do what I do and take my place"* (41).

### Affect School leaders' experiences

**Affect awareness is important in eating disorders.**   The AS leaders pointed out that patients with ED generally have difficulties with affect/emotion awareness, and that it is important for these patients to gain more knowledge of how their emotional systems function, such as how affects are manifested and experienced in the body. They considered that discussions, especially on negative affects were important since this became a kind of revelation, and thus a foundation for behavioural change. Avoidant behaviour was considered one of the main objectives of treatment, and as a risk- and maintenance factor for ED. By becoming aware of why you do what you do, it can become a key to changing your behaviour and initiate healing.

*"There are very many negative affects, and many that patients want to avoid, which may also be the cause of some problems, because you adopt behaviours when you want to escape from the ones that aggravate your problems. To actually talk about them is really important"* (AS leader B).

*"To some* (participants) *affects can be very new knowledge, what happens in the body, and how to separate different affects from each other, and I think that the Affect School can be somewhat normalising for participants"* (AS leader C).

However, the AS leaders also experienced the sessions as unpredictable, but in a positive way, regarding how participants reacted to different affects. Surprisingly, participants had difficulties with joy, which is the first affect in the AS, and usually considered an easy affect to talk about.

*"Affects that weren't thought to be particularly difficult aroused very strong feelings in participants. I've taken this to my other tasks"* (AS leader B).

**Group meetings create opportunities and challenges.**   The group format of the AS school was appreciated by the AS leaders. Although some groups were silent and required considerable activity from the leaders, the benefit of meeting participants in a group was perceived positively. Aspects such as "normalisation" when sharing experiences with others and "exposure" to one's own and others' feelings and experiences were regarded as significant. The opportunity to share feelings and life stories with others with ED was considered to increase acceptance for one's own emotional functioning.

*"Group treatment generally adds a lot that you don't get in individual therapy because you get the opportunity to share with others in a similar situation. There's great gain in taking part in other people's stories and seeing that you're not alone with your difficulties: perhaps you gain acceptance for the problems you have"* (AS leader A).

However, how well it worked was individual; some participants took up (too) much space, which sometimes impeded the other participants, while a few participants could not bear to hear about the difficult situation of others.

> *"It's really interesting, sometimes the groups work out well and sometimes you notice that 'oh, this wasn't good, that some dominate and others have a hard time adapting to it. Or, that the group doesn't really get on, and I mean that it's more than personality that rules"* (AS leader B).

Differences between group members could either be positive (e.g. participants that had preceded in their treatment could act as traction for participants with a shorter treatment history), or negative (e.g. participants with personality traits that were perceived as too different compared to the rest of the group). Eating disorder duration, personality traits, and co-morbidity were considered to obstruct a positive group process in some cases. Having different ED diagnoses was usually not a problem, and different degrees of recovery were mostly considered an aid and a hope for those who had a longer road to recovery.

> *"I experienced that it worked well to have different* (diagnoses) *in the groups where I was a leader. There are so many parameters that are similar that you may be able to see yourself. Everybody has such contempt for their bodies and thoughts about food so it's very reminiscent of each other, although expressed in slightly different ways"* (Therapist A).

It could also be the other way around; when more recovered participants listened to participants with more severe ED symptoms this could either strengthen their recovery process or trigger old symptoms. The setup of the AS allowed participants to refrain from plunging too deeply into in the discussions, which was experienced in some groups.

> *"Some participants presented milder examples so, so as not go into depth, while others shared more fully"* (AS leader A).

The AS leaders experienced that the group sizes for two of the groups were too small (three participants or two when someone was missing) and preferred groups of four to five. They feared that participants in undersized groups got insufficient input from others. Originally, the plan was to engage about eight participants in each group, but after a few smaller groups the AS leaders realised that listening to eight affect discussions in a row could be quite tiresome.

> *"I do not feel that it could have been eight people if we had the same setup, it would have been very, very boring for the participants to sit and listen and go through exactly the same questions every week without throwing them out a little, maybe put together, the most important thing may not be that everyone gets exactly the same questions"* (AS leader B).

## The affect school setup needs further customisation

Above all, the AS leaders were concerned about the setup and manual of the intervention. Clarity, structure and length of intervention were described as positive. However, suggestions came up on how to make, above all, the manual better. An experienced downside with the setup was that discussions easily became more like interviews and were experienced as irritating.

*"In fact, the manual has its pros and cons. I think the layout was somewhat rigid . . . the others were often very quiet when me and the other leader spoke, and it was difficult to get them engaged in the conversation"* (AS leader C).

Suggestions were to better adapt the content to suit participants with ED (although a small change was implemented into the manual) and to provide the possibility to weave in sections on ED into each affect. Discussions were suggested to be more open and provide a greater opportunity for shorter posts from participants rather than long "interviews". Some of the content was experienced as contradictory, with confusions of concepts e.g. a mix between affects and emotions despite definitions of the concepts. There was an expressed desire for a section on anxiety in the manual. Above all, a need was expressed for tools to better handle their ED, home assignments for participants, and to make the handouts more attractive. Participants dropped their AS work between sessions, which was considered negative. Another suggestion was that therapists outside the intervention should have access to the manual, and thus be able to enhance the intervention within individual treatment, or at least provide a meeting with the participant and psychotherapist regarding their work in the AS.

## Discussion

The AS participants described how the AS made them aware that all affects are important, and that even affects experienced as negative are vital, and should thus be experienced and expressed. Previous research has shown that people suffering from AN have difficulties in acknowledging emotions and experience discomfort when expressing negative emotions [18]. The AS leaders considered affect awareness important to patients with ED as avoidant, and sometimes detrimental, behaviour is connected to affects, which also has been shown previously [1, 3, 4, 19, 20]. The description of the aetiology, function, and utility of all affects presented in the AS seemed to help participants to allow themselves to experience and express affects/emotions without feeling ashamed, which in turn could increase empowerment and reduce self-criticism [21].

Despite positive narratives on perceived utility and permittance to experiencing and expressing affects, participants reported no experience of significant changes in their life situation, mood, or ED. Their overall picture was that the AS was part of a process, "a piece of a puzzle" leading to future results, but that change takes time. Another study of group treatment for inpatients with AN, showed that patients wished for more help for integrating new skills with practical life, such as more practical strategies or tools for change outside the group/ward [22]. This may indicate a gap between good intentions in treatment and transfer to life outside the group/ward, and perhaps the current intervention would have benefited from more tools and concrete home assignments between sessions for the participants, or therapist support for the ongoing AS work in the participants' parallel ED treatments. The AS model actually suggests a continuation of the group intervention with individual script analysis [10, 12], but this would require more education on the model for all treatment staff members at the unit and in the present study the AS leaders were the only staff members who had been trained in the intervention.

Despite that several of the participants had worried about participating in a group intervention, it ended with all of them feeling that the group in particular was a positive experience. Leaders and participants shared the view that the group format gave opportunities to experiences of recognition and normalisation of thoughts, behaviours, and feelings which were regarded helpful. Their narratives were in line with Yalom's [23] description of individuals' benefits of group processing as the possibility to experience universality, identification,

interpersonal learning, and installation of hope, which was supported by other interview studies on group therapy [22, 24]. Successful outcomes of psychological interventions have shown to be related to highly emotional and reflective states, e.g. less avoidance of negative affects during sessions [25]. One problem described by both the AS leaders and AS participants was that some participants "played it safe" and shared less demanding examples of affective experiences. This behaviour probably reduces the effectiveness of the treatment for the individual patient since it leads to a loss of considerable parts of the intervention, i.e. experiences of intense affects necessary for emotional and behavioural change [26] and it also counteracts beneficial group processes. A few participants described other participants' stories as painful, and that they usually took on their suffering. These descriptions were examples of adverse group reactions, which can naturally emerge in groups (especially with patients). It is important to notice and handle these in a professional manner [27]. Another problem raised by both participants and AS leaders was that when one participant had a more severe ED condition compared to the other participants and took more time discussing her ED behaviours, thus triggering another participant who had progressed further in the recovery process. This concern regarding social contagion in group therapy settings was highlighted by Vandereycken [28] and caution regarding this should be taken when enrolling patients to group interventions. Experiences of being less ill compared to other participants and experiencing that more severely ill participants are dominating discussions has also been described by Wannlass, Moreno, & Thomson [29]. However, in the present intervention the small group size allowed participants adequate time, and no participant complained about being given too little time, or of being stressed.

Although both participants' and AS leaders' found the intervention useful, the AS leaders strongly felt the handouts could have had a better lay-out, containing more on e.g. anxiety or connections between affects and the ED. Although they were positive to the content and structure, AS leaders were also critical to the individual affect interviews, which did not lead to discussion. However, the AS participants were satisfied and uncritical about both the structure and the handouts.

## Limitations

In the present study our interest was in how participants and leaders described their experiences of the AS. The authors were aware of, and considered the risk for, confusion between research and therapy when carrying out qualitative interviews [30]. The interviewers had no therapeutic/treatment relationship with the participants in the study. The second author had not worked with patients with ED thus providing the perspective of the "naïve inquirer" [31]. Another bias that may have occurred is that some narratives may have been guided by participants' wishes to display socially desirable behaviour, which implies a risk, not only in interview studies but in all studies where participants are aware of being studied. Although our sample consisted of women of differing ages and ED diagnoses to represent clinical reality, it was limited to ethnically Swedish women. As the male patients at the clinic at the time declined to participate in the AS study the male perspective is lacking, which is a shortcoming. A large part of the ED population is young, while participants in this study were 19–43 years old, and thus we do not know how younger patients would have responded to the intervention. The present study was undertaken as part of an RCT, and since researchers have experienced difficulties recruiting adult patients with ED to RCTs, recruitment of adolescent patients could be more feasible [32]. Due to consecutive patient enrolment, it was not possible to arrange diagnose-specific groups. Thus, participants in this study had different ED diagnoses, and it is possible that a group of patients with restrictive AN also would have reported more difficulties [33, 34].

The patients also received different types of parallel treatments since the present study aimed at a comparison between regular treatment, which varied, and AS as an additional treatment. There may also be objections to interviews as a research method. It has been argued that it is impossible to gain unbiased knowledge, since researchers always color results with pre-understanding [35]. The researchers in this study have tried to deal with this problem by using and communicating reflexivity [31].

## Conclusions

The results suggest that the Affect School provided an acceptance for experiencing all sorts of affects. Both leaders and participants considered working with affects necessary. At the time of the interviews, participants did not feel they had experienced significant changes in their life situation, mood, or ED. Instead, they considered AS to be a piece of the puzzle for a coming change. The results also suggest that being a part of a group, and thus sharing experiences as well as increasing affect awareness, was experienced as positive by participants and leaders, but the model could be further developed to better adapt to patients with an ED.

## Acknowledgments

The authors are grateful to the participants in this interview study and the assistance of the staff at the AnorexiBulimiCenter, Region Kalmar County, Sweden, for their support and help with inclusion of participants.

## Author Contributions

**Conceptualization:** Suzanne Petersson, Ingrid Wåhlin.

**Formal analysis:** Suzanne Petersson, Ingrid Wåhlin.

**Funding acquisition:** Suzanne Petersson.

**Investigation:** Suzanne Petersson, Ingrid Wåhlin.

**Methodology:** Suzanne Petersson, Ingrid Wåhlin.

**Project administration:** Suzanne Petersson.

**Validation:** Ingrid Wåhlin.

**Writing – original draft:** Suzanne Petersson, Ingrid Wåhlin.

**Writing – review & editing:** Suzanne Petersson, Ingrid Wåhlin.

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
