## [Decision Letter · Decision Letter 0]

22 Mar 2022

PONE-D-21-16822Patient and Psychologist Experiences of the Affect School as Additional Treatment in a Swedish Eating Disorder Unit: an Interview StudyPLOS ONE

Dear Dr. Petersson,

Thank you for submitting your manuscript to PLOS ONE. After careful consideration, we feel that it has merit but does not fully meet PLOS ONE’s publication criteria as it currently stands. Therefore, we invite you to submit a revised version of the manuscript that addresses the points raised during the review process.

The reviewers raised a number of concerns regarding the study design (and sample size), the high risk of bias and the reporting of results. Their comments can be viewed in full, below.

We look forward to receiving your revised manuscript.

Kind regards,

Natasha McDonald PhD

Associate Editor

PLOS ONE

Journal Requirements:

2. Thank you for including your ethics statement: "The study was conducted according to the principals of the Helsinki declaration. Prior to interviews participants were provided written and oral information about the study. It was emphasised that participation was voluntary, and that participation or refusal would not affect treatment. Participants were informed that presentation of the data would be handled with confidentiality so that no statement could be traced to any single informant. The study was approved by the research ethics committee of Linköping (2017/531-31). The interviewers had no treatment relationship to the interviewees." 

a) Please provide additional details regarding participant consent. In the ethics statement in the Methods and online submission information, please ensure that you have specified what type you obtained (for instance, written or verbal, and if verbal, how it was documented and witnessed). If your study included minors, state whether you obtained consent from parents or guardians. If the need for consent was waived by the ethics committee, please include this information.

Reviewers' comments:

Reviewer's Responses to Questions

**Comments to the Author**

1. Is the manuscript technically sound, and do the data support the conclusions?

Reviewer #1: Yes

Reviewer #2: Yes

2. Has the statistical analysis been performed appropriately and rigorously? 

Reviewer #1: Yes

Reviewer #2: No

3. Have the authors made all data underlying the findings in their manuscript fully available?

Reviewer #1: Yes

Reviewer #2: Yes

4. Is the manuscript presented in an intelligible fashion and written in standard English?

Reviewer #1: Yes

Reviewer #2: Yes

5. Review Comments to the Author

Reviewer #1: The findings reported in the manuscript entitled "Patient and Psychologist Experiences of the Affect School as Additional Treatment in a Swedish Eating Disorder Unit: an Interview Study" are an important contribution to the field of eating disorders and of high relevance to clinical practice. From this reviewer's perspective, I consider that the methodology employed was suitable to achieve the objective of the present study; however, the following suggestions are made for clarity and understanding

- It is mentioned that "Parallel to the intervention both groups received treatment as usual." It would be useful to know if the type of treatment the patients had, was the same or were there differences (approach, psychological and medical treatment, psychological treatment only).

- According to the manuscript, it seems that there is information on the length of time the patients have had the diagnosis of ED and its comorbidity, why not include them in the description of the sample?

- Why were the groups not formed according to diagnosis, i.e., one group of AN, another of BN, etc.?

- The discussion needs more work, as it is mainly a repetition of the results and little comparison with the existing literature.

Reviewer #2: This is a study of experiences from an "Affect school" in which patients with Eating Disorders were enrolled. The study included 9 individuals who were interviewed and the responses were analysed with Thematic Analysis. Nine themes were revealed at the analysis, six from the participants, which, according to the authors, amongst others revealed that the Affect School provided an acceptance for experiencing all sorts of affects.

There are some major issues with this study:

- The study included 9 individuals which, albeit being a qualitative study, is a very small sample.

- the is a high risk of bias in this study where both the affect school "training", the interrelations between the enrolled subjects, and the interviewers relation to the affect school and the interviewed subjects may all have influenced the responses the subjects provided.

- it cannot be ruled out that the experiences were, so to speak" manipulated" by the Affect school activities.

- some of the themes seem closely related e.g. Worries about group participation”, “Not alone anymore”, “Shared stories can also be painful”, “New insights about oneself and others”, “Relationships as a foundation for affective work”, which cast some doubt about the method for identifying themes.

- The conclusion that "the results showed" is very bold. This type of qualitative interview and analysis may at best provide som indication of an outcome such as "acceptance" and thereby the statement should be changed from showed to "suggest". This is also important in view of the fairly substantial bias present in this study.

- Page 5, Mehods: In "Study setting" a diagnostic method is described; this should be moved to a separate section under Methods

- Page 5, Mehods: It is not adequate clinical praxis to diagnose different eating disorders with EDE-q and EDI alone and thus, either add information on how the diagnoses were set, or, stress that the patients were not fully diagnosed.

- Results; after each theme, the authors should indicate how many of the subjects vs AS leaders stated these (e.g. 3 of 9)

- Discussion; the structure is unorthodox. The authors should start by presenting and discussing the main results, thereafter other findings, thereafter limitations and finally conclude. The whole discussion needs to be restructured and rewritten.

6. PLOS authors have the option to publish the peer review history of their article (what does this mean?). If published, this will include your full peer review and any attached files.

Reviewer #1: No

Reviewer #2: No

---

## [Author Response · Author response to Decision Letter 0]

4 May 2022

To the journal: Thank you for your comments. We wish to add “A Piece of a Puzzle” in the beginning of the title of the manuscript and hope that this will be alright. 

Journal Requirements:

2. Thank you for including your ethics statement: "The study was conducted according to the principals of the Helsinki declaration. Prior to interviews participants were provided written and oral information about the study. It was emphasised that participation was voluntary, and that participation or refusal would not affect treatment. Participants were informed that presentation of the data would be handled with confidentiality so that no statement could be traced to any single informant. The study was approved by the research ethics committee of Linköping (2017/531-31). The interviewers had no treatment relationship to the interviewees." 

a) Please provide additional details regarding participant consent. In the ethics statement in the Methods and online submission information, please ensure that you have specified what type you obtained (for instance, written or verbal, and if verbal, how it was documented and witnessed). If your study included minors, state whether you obtained consent from parents or guardians. If the need for consent was waived by the ethics committee, please include this information. Once you have amended this/these statement(s) in the Methods section of the manuscript, please add the same text to the “Ethics Statement” field of the submission form (via “Edit Submission”).

Answer: We have added: “All participants were adults (≥18 years old).” and “Participants signed an informed consent to participation which was attached to the written information. Participants kept one copy and the other copy was saved in a safe by the researchers” to the Ethical Approval and consent to participate section. We have also copied the text into the Ethics Statements in the online submission information.

Answer: This is now taken care of. 

Answer: Data Availability Statement: Data cannot be shared publicly because of ethical reasons. Data is private, confidential and of a sensitive nature and there is a risk of identification through the

interviews. Data can be requested from the corresponding author (contact suzanne.petersson@regionkalmar.se) for researchers who meet the criteria for access to confidential data. Region Kalmar County, as principal for this study, is chiefly responsible for the research data collected. 

Answer: Ethical approval and consent to participate is moved to the Methods section.

To the Reviewers:

Thank you very much for valuable and relevant comments. We have made efforts to meet your suggestions. Changes are marked with red text and commented below. We believe the manuscript has been considerably improved and hope that you now find it suitable for publication. 

Reviewers' comments:

Reviewer's Responses to Questions

Comments to the Author

1. Is the manuscript technically sound, and do the data support the conclusions?

Reviewer #1: Yes

Reviewer #2: Yes

2. Has the statistical analysis been performed appropriately and rigorously? 

Reviewer #1: Yes

Reviewer #2: No

3. Have the authors made all data underlying the findings in their manuscript fully available?

Reviewer #1: Yes

Reviewer #2: Yes

4. Is the manuscript presented in an intelligible fashion and written in standard English?

Reviewer #1: Yes

Reviewer #2: Yes

5. Review Comments to the Author

Reviewer #1: The findings reported in the manuscript entitled "Patient and Psychologist Experiences of the Affect School as Additional Treatment in a Swedish Eating Disorder Unit: an Interview Study" are an important contribution to the field of eating disorders and of high relevance to clinical practice. From this reviewer's perspective, I consider that the methodology employed was suitable to achieve the objective of the present study; however, the following suggestions are made for clarity and understanding

- It is mentioned that "Parallel to the intervention both groups received treatment as usual." It would be useful to know if the type of treatment the patients had, was the same or were there differences (approach, psychological and medical treatment, psychological treatment only).

Answer: They received different treatments, which we now have described in the text “which in these cases included individual psychotherapy (CBT) and in some cases also day care and/or physiotherapy”. We have also added “The patients also received different types of parallel treatments since the RCT aimed at a comparison between regular treatment, which was varying, and AS as additional treatment” to the Limitations section.

- According to the manuscript, it seems that there is information on the length of time the patients have had the diagnosis of ED and its comorbidity, why not include them in the description of the sample?

Answer: Unfortunately, we do not have reliable data on co-morbidity. But we have added the ED duration (self-referred). We also added this to the Participants description.

- Why were the groups not formed according to diagnosis, i.e., one group of AN, another of BN, etc.?

Answer: Participants were included consecutively and thus it was not possible to arrange groups with diagnose specific groups. This has now been added at the limitations section.

- The discussion needs more work, as it is mainly a repetition of the results and little comparison with the existing literature.

Answer: We have rewritten the Discussion section and hope that it will be more readable now (and with more relevant references).

Reviewer #2: This is a study of experiences from an "Affect school" in which patients with Eating Disorders were enrolled. The study included 9 individuals who were interviewed and the responses were analysed with Thematic Analysis. Nine themes were revealed at the analysis, six from the participants, which, according to the authors, amongst others revealed that the Affect School provided an acceptance for experiencing all sorts of affects.

There are some major issues with this study:

- The study included 9 individuals which, albeit being a qualitative study, is a very small sample.

Answer: Yes, the sample was 9 plus 3 AS leaders. The AS leaders were only three, thus we interviewed all of them. The number of participants in the AS group in the RCT were 21 and nine of them also agreed to participate in the interview study. In qualitative studies, you strive for a variation of parameters considered important (a broad representation) capturing different opions. The patients had different ED diagnoses and were of representative ages. The patients were also enrolled from five AS groups. Despite the small sample from different participants, the content was quite consistent which is why we considered the number of informants was sufficient. 

- the is a high risk of bias in this study where both the affect school "training", the interrelations between the enrolled subjects, and the interviewers relation to the affect school and the interviewed subjects may all have influenced the responses the subjects provided.

Answer: We (the authors) are not educated in the affect school and neither of us had any relation to the interviewed AS participants (the patients). Staff was interviewed by IW, who only had contact with them at the time of the interview. We have now clarified this under the Data collection section. 

- it cannot be ruled out that the experiences were, so to speak" manipulated" by the Affect school activities.

Answer: All questions were about experiences from the AS. But of course experiences and interpretations of situations always affect people's experiences, no one exists without their history and context. The possible effect of the AS was evaluated in another study.

- some of the themes seem closely related e.g. Worries about group participation”, “Not alone anymore”, “Shared stories can also be painful”, “New insights about oneself and others”, “Relationships as a foundation for affective work”, which cast some doubt about the method for identifying themes.

Answer: Time has gone (a year now) and we also think that two of the themes are related. Due to this we have re-analysed the material and combined “New insights about oneself and others” and “Not alone anymore”. We extended the name of the theme “Relationships as a foundation for affective work” into “Relationships outside the Affect School as a foundation for affective work” in order to clarify this theme. The reason for keeping “Shared stories can also be painful” is that this is a group experience which has received sparse attention in research on group interventions.

- The conclusion that "the results showed" is very bold. This type of qualitative interview and analysis may at best provide som indication of an outcome such as "acceptance" and thereby the statement should be changed from showed to "suggest". This is also important in view of the fairly substantial bias present in this study.

Answer: Yes, this is certainly correct and we have changed the word into “suggested” (as suggested).

- Page 5, Mehods: In "Study setting" a diagnostic method is described; this should be moved to a separate section under Methods

Answer: This is now changed according to your advice.

- Page 5, Mehods: It is not adequate clinical praxis to diagnose different eating disorders with EDE-q and EDI alone and thus, either add information on how the diagnoses were set, or, stress that the patients were not fully diagnosed.

Answer: No that is right. The patients were diagnosed with the Structured Eating Disorder Interview (SEDI) which was adapted to DSM-5 and not the EDI. We have added the abbreviation of the Structured Eating Disorder Interview to the text and also added that the diagnoses were set by experienced psychologists and psychotherapists, which was the case. Regarding the EDEQ, this was not a foundation for diagnosis, so we have removed this from the text.

- Results; after each theme, the authors should indicate how many of the subjects vs AS leaders stated these (e.g. 3 of 9)

Answer: Theme frequency is not to recommended in TA (Braun & Clarke, 2022)! There are many reasons not to use numbers in qualitative analysis e.g. that “more should be better”, that “numbers are better”, each data item is not directly comparable to another (it is not possible to know if those who did not mention the theme thought that the theme was not relevant).

- Discussion; the structure is unorthodox. The authors should start by presenting and discussing the main results, thereafter other findings, thereafter limitations and finally conclude. The whole discussion needs to be restructured and rewritten.

Answer: The sections and headings were in this order, but the Conclusion heading was missing, we apologize for this and have now completed the manuscript. We have also shortened and restructured the Discussion section in order to make it more readable.

---

## [Decision Letter · Decision Letter 1]

11 Jul 2022

A Piece of a Puzzle - Patient and Psychologist Experiences of the Affect School as Additional Treatment in a Swedish Eating Disorder Unit

PONE-D-21-16822R1

Dear Dr. Petersson,

We’re pleased to inform you that your manuscript has been judged scientifically suitable for publication and will be formally accepted for publication once it meets all outstanding technical requirements.

Kind regards,

George Vousden

Staff Editor

PLOS ONE

Additional Editor Comments (optional):

Reviewers' comments:

Reviewer's Responses to Questions

**Comments to the Author**

1. If the authors have adequately addressed your comments raised in a previous round of review and you feel that this manuscript is now acceptable for publication, you may indicate that here to bypass the “Comments to the Author” section, enter your conflict of interest statement in the “Confidential to Editor” section, and submit your "Accept" recommendation.

Reviewer #1: All comments have been addressed

Reviewer #3: (No Response)

2. Is the manuscript technically sound, and do the data support the conclusions?

Reviewer #1: (No Response)

Reviewer #3: Yes

3. Has the statistical analysis been performed appropriately and rigorously? 

Reviewer #1: (No Response)

Reviewer #3: N/A

4. Have the authors made all data underlying the findings in their manuscript fully available?

Reviewer #1: (No Response)

Reviewer #3: Yes

5. Is the manuscript presented in an intelligible fashion and written in standard English?

Reviewer #1: (No Response)

Reviewer #3: Yes

6. Review Comments to the Author

Reviewer #1: (No Response)

Reviewer #3: The authors describe an interesting initial qualitative evaluation of an affect focused intervention (the Affect School) adapted for eating disorders. As noted by Reviewer 2 in the initial review of the manuscript, the study has limitations (such as the small sample size and lack of quantitative data), but these are appropriately acknowledged by the authors in the discussion section. The analysis techniques used are appropriate for qualitative data. The authors appear to offer a nuanced presentation of the data, noting both strengths and potential shortcomings of the Affect School intervention. The inferences that can be drawn from this study alone are limited, but as a pilot study, I believe the information presented in this article will be useful for the general scientific and clinical community to continue refining group affect focused interventions for people with eating disorders.

7. PLOS authors have the option to publish the peer review history of their article (what does this mean?). If published, this will include your full peer review and any attached files.

Reviewer #1: No

Reviewer #3: No

---

## [Editor Report · Acceptance letter]

18 Jul 2022

PONE-D-21-16822R1 

A Piece of a Puzzle - Patient and Psychologist Experiences of the Affect School as Additional Treatment in a Swedish Eating Disorder Unit 

Dear Dr. Petersson:

I'm pleased to inform you that your manuscript has been deemed suitable for publication in PLOS ONE. Congratulations! Your manuscript is now with our production department. 

Kind regards, 

on behalf of

Dr. George Vousden 

Staff Editor

PLOS ONE